# Prevalence and correlates of malaria and undernutrition among acutely febrile children visiting temporary malaria screening sites in war-torn areas of Northeast Ethiopia

**Habtu Debash**◉*, **Ermiyas Alemayehu**◉, **Melaku Ashagrie Belete**◉, **Hussen Ebrahim**, **Ousman Mohammed, Daniel Gebretsadik, Mihret Tilahun, Agumas Shibabaw, Zewudu Mulatie**◉, **Bruktawit Eshetu, Saba Gebremichael, Alemu Gedefie**◉

Department of Medical Laboratory Sciences, College of Medicine and Health Sciences, Wollo University, Dessie, Ethiopia

* habtudebash@gmail.com

## Abstract

### Background

Malaria and undernutrition pose challenges for children in conflict-affected areas. Understanding the prevalence and risk factors for these issues in war-torn communities is important to effectively design aid efforts and select interventions. This study aimed to determine the prevalence and correlates of malaria and undernutrition among febrile children in northeast Ethiopia to help address these problems.

### Methods

A cross-sectional study was conducted from November 2022 to January 2023. Four hundred twenty-two children were enrolled using a systematic random sampling technique. Data on associated factor variables were collected via questionnaire. Capillary blood samples were collected from each child to prepare thick and thin blood films, which were stained with Giemsa and examined microscopically. Height and weight measurements were also taken using a meter and a standard calibrated balance. The data were analyzed in SPSS 26.0 using bivariable and multivariable logistic regression to determine associations between correlates, malaria infection, and undernutrition. Statistical significance was set at p < 0.05.

### Results

The overall malaria prevalence among screened children at temporary sites in the conflict-affected areas of Northeast Ethiopia was 65.9% (278/422). *Plasmodium falciparum*, *P. vivax*, and mixed infections accounted for 74.1%, 19.8%, and 6.1% of the cases, respectively. The presence of stagnant water (P<0.01), improper utilization of ITN, inadequate number of ITNs per family size (P<0.01), and absence of environmental management (P<0.01) were independent predictors of malaria. On the other hand, the overall

**Data Availability Statement:** All data generated in this study are included in the manuscript.

**Funding:** The author(s) received no specific funding for this work.

**Competing interests:** The authors have declared that no competing interests exist.

undernutrition prevalence was 54.7% (231/422), including 26.5% underweight, 16.8% stunted, and 11.4% wasted. Dietary diversity score (P<0.01), meal frequency (P<0.01), and confirmed malaria infection (P<0.01) were significantly associated with underlnutrition.

## Conclusion

Both malaria and undernutrition burdens were high among the children in this study. The findings suggest combined prevention measures for malaria and undernutrition should be strengthened in this region.

## Introduction

Armed conflicts and regional struggles influence disease emergence by disrupting transmission through infrastructure destruction and healthcare system collapse [1]. Conflict in Ethiopia over a year caused massive displacement, with thousands dead and over 2 million displaced from their homes. More than 542,000 people fled unrest within the Amhara region, severely impacting the Waghemra zone amid the region's turmoil [2]. Armed conflicts negatively impact healthcare availability and accessibility by widespread devastation of institutions, overburdening precarious systems [3]. Consequences include destroyed facilities, killed workers, malnutrition/harm to children, increased mental health issues, epidemics, and other public health problems [3].

The conflict halted prior anti-malarial efforts, allowing potential epidemics in high-transmission areas if health systems falter, treatments are interrupted, or non-immune groups migrate [4]. When the anti-malarial program ended, malaria re-emerged, turning the territory into an endemic area [5]. This could contribute to the spread of drug resistance, jeopardizing long-term elimination goals [6]. Conflict areas are prone to malaria infections due to lack of immunity, concentrated exposed populations, limited insecticide-treated Bed nets (ITNs) and Indoor residual Spray (IRS), and insufficient rapid diagnosis, treatment, and responses [7, 8]. Other risk factors include nighttime activities outside, wearing shorts, living in unfinished buildings, inadequate drainage, and acute malnutrition in children [9, 10].

Combating malnutrition is especially challenging in conflict zones since women and children are disproportionately impacted. Increased global hunger was primarily driven by conflicts, explaining why food insecurity and malnutrition were most severe in these conflict-affected areas. [11–13]. Conflicts negatively influence food security by causing large displacements, deep economic recessions, rising inflation and unemployment, and eroding funds for social safety and healthcare. Conflicts disrupt food production, harvesting, processing, transportation, and marketing in areas where agriculture is the primary source of income, resulting in low resilience. As a result, worsening food insecurity might heighten tensions and the risks of additional conflicts [11, 12].

People in conflict zones face higher malnutrition risks [7]. Research shows factors like age, gender, education, and occupation influence nutrition [14, 15]. Moreover, low birth weight, immunization, maternal nutrition, and food intake impact childhood malnutrition [16, 17]. The relationship between malaria and nutrition is complex, as each can increase the risks of other [18]. Over a year of conflict in Northeast Ethiopia seriously weakened its healthcare system and disrupted services, making communities highly vulnerable to interconnected issues like malaria and malnutrition.

Understanding the magnitude and key risk factors for malaria infections and undernutrition in war-affected communities is critical for designing humanitarian responses and selecting appropriate intervention strategies for under nutrition and malaria prevention, control, and even elimination. This is only possible with adequate research and evidence to support the development of effective and sustainable management strategies. The purpose of this study was to provide an overview of the prevalence of malaria infections, undernutrition, and local context correlates among acutely febrile children seeking health care at temporary malaria screening sites in war-torn areas of Northeast Ethiopia.

## Methods

### Study area, design, and period

A descriptive cross-sectional survey was conducted on acutely febrile children receiving healthcare at temporary malaria screening sites from November 1st, 2022, to January 30th, 2023, in Waghemra Zone, Northeast Ethiopia. Waghemra Zone is one of eleven zones in Ethiopia's Amhara Region. The Waghemra zone is defined by the following latitude and longitude coordinates: 12˚ 45' 54" N, 38˚ 50' 34.8" E. The study site is located in the dissected landscapes of Waghemra Zone's Tekezie Basin, where natural and man-made calamities such as violence, migration, and food insecurity have posed severe challenges since 2021/22. This zone is divided into eight districts with a total population of 536,129 people. This sero-survey was conducted in Abergele Woreda where children came for treatment at temporary malaria screening and treatment sites. However, some kebeles in the Abergele District were excluded from the data collection due to the ongoing war in the area. In addition to the conflict, the climate and terrain of the region provide suitable breeding grounds for Anopheles mosquitoes.

### Sample size determination

The sample size was calculated using a single population proportion formula, by taking 95% confidence limits (Zα/2 = 1.96), and 5% margin of error (d) with a maximum proportion of 50%. So, the sample size = $Z(\alpha/2)^2 P (1-P)/d^2$, $(1.96))^2 * 0.221 (1-0.221)/(0.05)^2 = 384$. Considering a 10% non-response rate, a total of 422 children were ultimately included in the study.

### Sampling techniques and procedure

A stratified sampling method was used to select the representative sample size. The Waghemra Zone health department had established rapid malaria response teams that provide malaria diagnosis and treatment at sites in Saka, Debrebirhan, and Beltarf. The total number of malaria-diagnosed febrile patients in the two months prior to the study was used as the sampling frame for each diagnostic and treatment center. Based on this, the total sample size was proportionally allocated such that 212, 124, and 86 febrile children were included from the Saka, Debrebirhan, and Beltarf, sites respectively. Systematic random sampling was applied to select study participants from each center.

### Inclusion and exclusion criteria

Children who presented with a fever ≥37.5˚C and were clinically suspected of having malaria were referred to the laboratory for a blood film examination. Children's parents who agreed to participate were included in the study. On the other hand, children who had taken anti-malarial drugs within 4 weeks before data collection were excluded.

## Data collection methods

**Socio-demographic and other factors.**   To collect information from each child, a pre-tested questionnaire was used, which included related elements such as background characteristics, behaviors, environmental, and other correlates important for malaria infection and undernutrition (S1 File). Parents or guardians of the children were interviewed about socio-demographic parameters and other associated factors. These factors include availability and utilization of insecticide-treated net (ITN), indoor residual spraying (IRS), presence of stagnant water, nighttime outdoor stays, and past malaria health information. All socio-demographic, behavioral and environmental correlates of malaria were prepared and reviewed by the investigative team, including those involved in data collection. All information on the questionnaire was standardized and adapted from the national malaria survey of Ethiopia and other published research in reputable journals. The questionnaire was developed in English and was translated into the local language (Amharic).

**Blood sample collection, processing, and examination.**   Following the interview, blood samples were aseptically collected from each child using a disposable sterile lancet via a finger prick. Approximately 6 μl and 2 μl of blood were used to prepare thick and thin smears, respectively. Both thin and thick blood films were prepared and air-dried. The thin blood film was then fixed with methanol and allowed to air dry. Thin and thick blood films were stained with 10% Giemsa staining solution for 10 minutes. After being rinsed with distilled water, the slides were air dried. Finally, the dried slides were examined by laboratory technologists at the health facilities. Thick smears were used to detect *Plasmodium* infections, while thin smears were used to identify parasite species using an oil immersion objective.

**Anthropometric measurements.**   Clinical nurses measured each child's weight and height using a digital scale that measures to the nearest 0.1 kilograms and a vertical meter that measures to the nearest 0.1 cm, respectively. Before measurement, the participant removed their shoes, jacket, and hair clips and positioned their feet together flat on the ground, with heels touching the back plate of the measuring instrument, legs straight, buttocks against the backboard, scapulae against the backboard, and arms at their sides. The study participants' weight and height were measured twice by separate nurses to decrease subjective error, and the mean measurement result was reported. The WHO AnthroPlus software was used to estimate undernutrition among schoolchildren. The WHO AnthroPlus software was used to calculate z-scores of height-for-age (HAZ) and body mass index-for-age z-score (BAZ). Children with a z-score < -2SD were classified as underweight (WHZ < -2SD), stunted (HAZ < -2SD), and wasted (BAZ < -2SD) [19].

*Dietary diversity score*. Dietary consumption in the 24 hours prior to the survey was assessed using questionnaires developed based on Food and Agriculture Organization (FAO) guidelines [20]. Consumed foods were classified into 10 food groups [21]. Consumption from each food group was reported, and one point was given if the food was consumed at least once in the last 24 hours and zero points if not consumed within that period. Therefore, the dietary diversity score (DDS) ranged from 0 to 10, and participants with a DDS of ≤3 were classified as poor diversity, whereas participants with a DDS of 4–6 and >6 were classified as middle and high diversity respectively [21].

## Quality control

The data collectors underwent training, and the questionnaire was pre-tested. To ensure the quality and integrity of the Giemsa staining, positive and negative blood film samples were taken. The microscopic slides were reviewed individually by two qualified laboratory technicians. Before being concluded as negative, hundreds of microscopic fields of the thick smear

were inspected. Any disparities between the findings of the first and second technicians were resolved by a third senior laboratory technologist.

## Statistical analysis

The gathered information was entered into Epi Data Version 4.2 and then exported to SPSS Version 26 for analysis. Narration and tabulation were used to present the descriptive statistics. Bivariate and multivariate logistic regression analyses were also used to assess the association between dependent and independent variables. All variables were analyzed individually for an association with malaria infection and undernutrition using bivariate logistic regression as random effects to account for correlation. A variable that had a P-value < 0.25 in the bivariate analysis was selected as a candidate for transfer to multivariate logistic regression analysis to control for confounding variables. A forward stepwise method was used, retaining any risk factors showing an association. During multivariate analysis, a P-value <0.05 at the 95% confidence interval was considered statistically significant.

## Ethical approval and consent to participate

Ethical clearance was obtained from the Ethical Review Committee of the College of Medicine and Health Sciences at Wollo University on the date 21/10/2022 with protocol number CMHS/126/2022. Permission was also obtained from the Waghemra Zone Health Office. After briefly describing the significance of the study to the children's parents or guardians, informed written consent was obtained from them. Confidentiality of the data was maintained. Finally, participants who were infected with the *Plasmodium* parasite received antimalarial treatment according to national malaria treatment guidelines. Those with undernutrition were referred to Tefera Hailu Memorial Hospital for further investigation. All methods were carried out in accordance with the Declaration of Helsinki guidelines and regulations.

## Results

### Socio-demographic characteristics of study participants

A total of 422 children participated in the study, with a 100% response rate. Of these, 238 (56.4%) were females and 184 (43.6%) were males. In terms of age, the highest number 176 (41.7%) were between 11 and 15 years. Similarly, the higher proportion (68.2%) of the children were from households with five or more family members, and 53.8% had never received a formal education and hence could not read or write. Furthermore, regarding testing sites, the highest proportion of participants (50.2%) was from the Saka testing site (Table 1).

### *Plasmodium* parasite distribution with socio-demographic variables

The overall prevalence of malaria in this study was 65.9% (278/422) (95% CI: 61.0–70.3). Among the malaria positive cases, the dominant *Plasmodium* species was *P. falciparum* in 206 cases (74.1%), followed by *P. vivax* in 55 cases (19.8%), and there were 17 cases (6.1%) with mixed infections of *P. falciparum* and *P. vivax*. Of the total cases, the prevalence of malaria was 156 (56.1%) in males and 122 (43.9%) in females. The difference in malaria prevalence by sex was not statistically significant (P = 0.87). Among the *Plasmodium* parasite-infected children, younger children had a lower prevalence of the *Plasmodium* parasite, with no statistically significant difference (P = 0.17). Furthermore, there was a higher prevalence of malaria parasites at the Saka testing site (78.3%) compared to the D/Brihan (72.8%) and Beltarf (62.2%) testing sites. The difference in prevalence between the different testing sites was statistically significant (P = 0.01) (Table 2).

**Table 1. Socio-demographic characteristics of children visiting temporal malaria screening sites in the war-torn areas of Northeast Ethiopia, 2023.**

| Variables | Alternatives | Number (%) |
|---|---|---|
| Sex | Male | 238 (56.4) |
| | Female | 184 (43.6) |
| Age (years) | ≤5 | 114 (27.0) |
| | 5–10 | 132 (31.3) |
| | 11–15 | 176 (41.7) |
| Residence | Rural | 376 (89.1) |
| | Urban | 46 (10.9) |
| Highest level of education of the parents/guardians | No school | 227 (53.8) |
| | Elementary school | 119 (28.2) |
| | High school | 48 (11.4) |
| | Higher education | 28 (6.6) |
| Parents occupation | Government employee | 18 (4.3) |
| | Merchant | 52 (12.3) |
| | Farmer | 352 (83.4) |
| Family size | ≥5 members | 288 (68.2) |
| | <5 members | 134 (21.8) |
| Testing site | Saka | 212 (50.2) |
| | Debre-brihan | 124 (29.4) |
| | Beltarf | 86 (20.4) |

**Table 2. Socio-demographic characteristics and malaria infection of children visiting temporal malaria screening sites in the war-torn areas of Northeast Ethiopia, 2023.**

| Variables | Alternatives | Positive n(%) | P. falciparum n (%) | P. vivax n(%) | Mixed n(%) | Negative n(%) | P-value |
|---|---|---|---|---|---|---|---|
| Sex | Male | 156 (65.5) | 112 (71.8) | 36 (23.1) | 8 (5.1) | 82 (34.5) | 0.871 |
| | Female | 122 (66.3) | 94 (77.0) | 19 (15.6) | 9 (7.4) | 62 (33.7) | |
| Age (years) | ≤5 | 70 (61.4) | 50 (71.4) | 13 (18.6) | 7 (10.0) | 44 (38.6) | 0.165 |
| | 6–10 | 98 (74.2) | 74 (75.5) | 18 (18.4) | 6 (8.1) | 34 (25.8) | |
| | 11–15 | 110 (62.5) | 82 (74.5) | 24 (21.8) | 4 (3.6) | 66 (37.5) | |
| Residence | Rural | 251 (66.8) | 184 (73.3) | 52 (20.7) | 15 (6.0) | 125 (33.2) | 0.276 |
| | Urban | 27 (58.7) | 22 (81.5) | 3 (11.1) | 2 (7.4) | 19 (41.7) | |
| Highest level of education of the parents | Illiterate | 150 (66.1) | 113 (75.3) | 29 (19.3) | 8 (5.3) | 77 (33.9) | 0.469 |
| | Elementary | 74 (62.2) | 56 (75.7) | 16 (21.6) | 2 (2.7) | 45 (37.8) | |
| | High school | 36 (75.0) | 25 (69.4) | 6 (16.7) | 5 (13.9) | 12 (25.0) | |
| | Higher education | 18 (64.3) | 12 (66.7) | 4 (22.2) | 2 (11.1) | 10 (35.7) | |
| Occupation | Gov.t employee | 13 (72.2) | 10 (76.9) | 3 (23.1) | 0 (0.0) | 5 (27.8) | |
| | Merchant | 39 (75.0) | 29 (74.4) | 8 (20.5) | 2 (5.1) | 13 (25.0) | |
| | Farmer | 226 (64.2) | 167 (73.9) | 44 (19.5) | 15 (6.6) | 126 (35.8) | 0.261 |
| Family size | ≥5 | 194 (67.4) | 141 (72.7) | 41 (21.1) | 12 (6.2) | 94 (32.6) | 0.346 |
| | <5 | 84 (62.7) | 65 (77.4) | 14 (16.7) | 5 (6.0) | 50 (37.3) | |
| Testing site | Saka | 152 (71.7) | 119 (78.3) | 25 (16.4) | 8 (5.3) | 60 (28.3) | 0.006 |
| | Debre-brihan | 81 (65.3) | 59 (72.8) | 18 (22.2) | 4 (4.9) | 43 (34.7) | |
| | Beltarf | 45 (52.3) | 28 (62.2) | 12 (26.7) | 5 (11.1) | 41 (47.7) | |
| **Total** | | 278 (65.9) | 206 (74.1) | 55 (19.8) | 17 (6.1) | 144 (34.1) | |

**Table 3. Binary logistic regression analysis of factors associated with malaria in the war-torn areas of Northeast Ethiopia, 2023.**

| Variables | Alternatives | Malaria status | | COR (95%CI) | P-value | AOR (95%CI) | P-value |
|---|---|---|---|---|---|---|---|
| | | Positive n (%) | Negative n (%) | | | | |
| Sex | Male | 156 (65.5) | 82 (34.5) | 0.967(.644–1.451) | 0.87 | | |
| | Female | 122 (66.3) | 62 (33.7) | 1 | | | |
| Age | ≤5 | 70 (61.4) | 44 (38.6) | 0.955(0.588–1.550) | 0.85 | | |
| | 6–10 | 98 (74.2) | 34 (25.8) | 0.552(0.321–0.950) | 0.30 | | |
| | 11–15 | 110 (62.5) | 66 (37.5) | 1 | | | |
| Residence | Rural | 251 (66.8) | 125 (33.2) | 0.708(379–1.322) | 0.28 | | |
| | Urban | 27 (58.7) | 19 (41.7) | 1 | | | |
| History of malaria | Yes | 99 (69.7) | 43 (30.3) | 1.299(.842–2.003) | 0.24* | 1.008(0.579–1.754) | 0.98 |
| | No | 179 (63.9) | 101 (36.1) | 1 | | | |
| Stagnant water near to the house | Yes | 188 (80.3) | 46 (19.7) | 4.450(2.892–6.849) | <0.01* | 3.225(1.987–5.234) | <0.01* |
| | No | 90 (47.9) | 98 (52.1) | 1 | | | |
| Proper utilization of ITNs | Yes | 90 (48.9) | 94 (51.1) | 1 | | | |
| | No | 188 (79.0) | 50 (21.0) | 3.927(2.567–6.008) | <0.01* | 3.701(2.233–6.135) | <0.01* |
| Number of ITNs | Adequate | 49 (43.4) | 64 (56.6) | | | | |
| | Inadequate | 229 (74.1) | 80 (25.9) | 3.739(2.382–5.868) | <0.01* | 3.561(2.089–6.070) | <0.01* |
| Outdoor stay at night | Yes | 62 (81.6) | 14 (18.4) | 2.665(1.435–4.952) | <0.01* | 1.082(0.492–2.380) | 0.84 |
| | No | 216 (62.4) | 130 (37.6) | 1 | | | |
| Environmental management | Yes | 42 (43.3) | 55 (56.7) | 1 | | | |
| | No | 236 (72.6) | 89 (27.4) | 3.472(2.171–5.555) | 0.0001* | 3.315(1.900–5.784) | <0.01* |
| Knowledge of care giver on malaria prevention | Good | 87 (54.4) | 73 (45.6) | 1 | | | |
| | Poor | 191 (72.9) | 71 (27.1) | 2.257(1.493–3.413) | 0.0001* | 1.859(0.739–3.034) | 0.48 |
| Nutritional status | Under nutrition | 170 (73.6) | 61 (26.4) | 2.142(1.423–3.225) | 0.0001* | 2.020(0.843–3.283) | 0.31 |
| | Normal | 108 (56.5) | 83 (43.5) | 1 | | 1 | |

*Significantly associated variables in bivariabel and multivariable analysis, AOR: Adjusted Odd Ratio, COR: Crud Odd Ratio, ITNs: Insecticide Treated Bed nets

## Factors associated with malaria infection

The presence of stagnant water, improper utilization of ITNs, an inadequate number of ITNs per family size, and the absence of environmental management were independent predictors of malaria. Participants who lived near mosquito breeding sites were more likely to have positive test results for malaria infections (adjusted odds ratio (AOR = 3.225; 95% CI: 1.987–5.234, P<0.01) compared to those who lived farther from mosquito breeding sites. Children who did not use ITNs were 3.701 times more likely to be infected with the *Plasmodium* parasite than those who did use ITNs (AOR = 3.701; 95% CI: 2.233–6.135, P<0.01). Likewise, the odds of contracting malaria among respondents who lived in households with an inadequate number of ITNs per family size were about three times higher than their counterparts (AOR: 3.561; 95% CI: 2.089–6.070, P<0.01). Furthermore, those who did not control their living environment were three times more likely to be at risk of contracting malaria (AOR = 3.315, 95% CI: 1.900–5.784, P<0.001) (Table 3).

## Nutritional status and associated factors

The overall prevalence of undernutrition among the children was 54.7% (231/422) (50.3–59.1). Of this, the prevalences of underweight, stunting, and wasting were 26.5% (22.3–30.9), 16.8% (13.3–20.6), and 11.4% (8.2–14.5), respectively (Table 4).

**Table 4. The prevalence of undernutrition among children in the war-torn areas of Northeast Ethiopia, 2023.**

| Nutritional status | Categories | | Frequency (n) | Percentage (%) |
|---|---|---|---|---|
| | Undernutrition | Wasted | 48 | 11.4 |
| | | Stunted | 71 | 16.8 |
| | | Underweight | 112 | 26.5 |
| | | Overall | 231 | 54.7% |
| | Normal | | 191 | 45.3% |

In the current study, the majority (72.7%) of the undernourished children were from families with five or more members. Similarly, 130 (56.3%) cases had caregivers who did not have any formal education. The binary logistic regression model showed that study participants who had a dietary diversity score (DDS) ≤ 3 (AOR = 2.147, 95% CI: 1.374–3.355, P<0.01), meal frequency of at most 3 times per day (AOR = 7.484, 95% CI: 4.740–11.817, P<0.01), and confirmed malaria infection (AOR = 2.215, 95% CI: 1.387–3.540, P<0.01) were strongly associated with undernutrition (Table 5).

## Discussion

This present study showed that malaria is a public health problem among acutely febrile children visiting temporary malaria screening sites in the war-torn areas of northeast Ethiopia. The study revealed that the overall malaria prevalence was 65.9% (95% CI: 61.0–70.3). This finding was consistent with a study done in Nigeria (77.6%) [22]. While Nigeria has a different malaria transmission context than Ethiopia, this previous study lends support to the magnitude of malaria burden identified, given the common challenges both countries face with malaria as an endemic disease.

In contrast, this result was higher than studies that found a 14.7% prevalence in the South Gondar Zone of northwest Ethiopia [23], 43.8% in the Jawi district of northwest Ethiopia [24], 20.5% in the East Shewa Zone of Ethiopia [25], 22.1% in Arba Minch Zuria District in southern Ethiopia [26], 58.2% in Anambra State, Nigeria [27], 55.3% in Niger [18], and 36.3% in Tanzania [28]. The high prevalence of malaria in this study might be related to the internal conflict in the study area over the past two years. This has resulted in deterioration of the health system and interruption of malaria prevention measures that previously kept malaria under control. This difference may also be due to seasonal variation, as our study was conducted during a major malaria transmission season, contrary to the other studies. Furthermore, the ecological niche for mosquito breeding could explain discrepancies in malaria prevalence between our study and others.

This study revealed that the predominant *Plasmodium* species detected in the blood of the children was *P. falciparum* in 206 cases (74.1%), followed by *P. vivax* in 55 cases (19.8%), and the remaining 17 cases (6.1%) showed mixed infections. This result disagreed with previous studies done in Ethiopia [23–26] and the national *Plasmodium* species prevalence, which indicated *P. falciparum* and *P. vivax* accounted for 60 and 40% of the malaria cases in the country, respectively. The higher dominance of *P. falciparum* over *P. vivax* is related to *P. falciparum*'s high proliferation in the host cell, the parasite's ability to infect all ages of red blood cells, and the parasite's treatment resistance [29]. Moreover, 6.1% of the children had mixed infections with *P. falciparum* and *P. vivax*. The reason for the mixed infections may be due to simultaneous infection by both species.

Malaria infections are frequently complicated and multifactorial, influenced by both natural and man-made causes [30]. Those children living within a 1 km radius of a mosquito breeding

**Table 5. Binary logistic regression analysis of potential factors associated with undernutrition among children in the war-torn areas of Northeast Ethiopia, 2023.**

| Variables | Categories | Undernutrition | | COR (95% CI) | P-value | AOR(95% CI) | P-value |
|---|---|---|---|---|---|---|---|
| | | **Yes** | **No** | | | | |
| | | **Number (%)** | **Number (%)** | | | | |
| Age (years) | ≤5 | 63 (55.3) | 51 (44.7) | 1.353(0.843–2.171) | 0.28 | | |
| | 6–10 | 84 (63.6) | 48 (36.4) | 0.706(0.423–1.178) | 0.26 | | |
| | 11–15 | 84 (47.7) | 92 (52.3) | 1 | | | |
| Residence | Rural | 203 (54.0) | 173 (46.0) | 0.754(0.403–1.411) | 0.38 | | |
| | Urban | 28 (60.9) | 18 (39.1) | 1 | | | |
| Highest level of education of the parents/ guardians | Unable to write and read | 130 (57.3) | 97 (42.7) | 0.536(0.227–1.268) | 0.36 | | |
| | Elementary school | 57 (47.9) | 62 (52.1) | 1.340(0.718–2.501) | 0.46 | | |
| | High school | 24 (50.0) | 24 (50.0) | 1.458(0.934–2.276) | 0.29 | | |
| | Higher education | 20 (71.4) | 8 (28.6) | 1 | | | |
| Family size | ≥5 | 168 (58.3) | 120 (41.7) | 1.578(1.045–2.383) | 0.03* | 1.426(0.887–2.292) | 1.43 |
| | <5 | 63 (47.0) | 71 (53.0) | 1 | | 1 | |
| Meal frequency per day | ≤3 | 154 (79.4) | 40 (20.6) | 7.55(4.848–11.759) | <0.01* | 7.484(4.740–11.817) | <0.01* |
| | >3 | 77 (33.8) | 151 (66.2) | 1 | | | |
| Dietary diversity score (DDS) | ≤3 | 137 (63.1) | 80 (36.9) | 2.022(1.370–2.985) | <0.01* | 2.147(1.374–3.355) | <0.01* |
| | >3 | 94 (45.9) | 111 (54.1) | 1 | | 1 | |
| Presence of diarrhoea/vomiting | Yes | 144 (59.5) | 98 (40.5) | 1.571(1.065–2.318) | 0.02* | 1.339(0.853–2.100) | 0.21 |
| | No | 87 (48.3) | 93 (51.7) | 1 | | 1 | |
| Malaria | Infected | 170 (61.2) | 108 (38.8) | 2.142(1.423–3.225) | <0.01* | 2.215(1.387–3.540) | <0.01* |
| | Not infected | 61 (42.4) | 83 (57.6) | 1 | | 1 | |

*Significantly associated variables in bivariabel and multivariable analysis, AOR: Adjusted Odd Ratio, COR: Crud Odd Ratio

site were three times as likely to develop malaria compared to those living in mosquito-free breeding sites. Stagnant water created by heavy rains provides an ideal breeding environment for mosquitoes and contributes to malaria epidemics [31, 32]. A similar conclusion was reached in research conducted in Simada district of northwest Ethiopia [32], and Waghemra Zone of northeast Ethiopia [33], which found an association between living near such water sources and contracting malaria.

This study also found that households that did not regularly use ITNs were 3.701 times more likely than those who used them regularly to be infected with the malaria parasite. This result is supported by similar previous studies conducted in the South Gondar and Arba Minch Zuria districts of Ethiopia [23, 26]. Insecticide-treated nets are an effective vector control method for preventing malaria transmission. When they are used consistently, the risk of getting a mosquito bite might be decreased [34, 35]. As a result, in malaria-endemic areas, local authorities should enforce the regular use of ITNs.

Children who lived in households with an inadequate number of ITNs were 3.561 times more likely to develop malaria. Inadequate ITNs in households might enforce infrequent utilization of ITNs in the community. It is recommended that there be at least one insecticide-treated net (ITN) for every two persons in the household. Furthermore, those children living in poorly controlled environments were three times at higher risk of contracting the disease. This finding is supported by a report from the Simada District in northwest Ethiopia [32]. Hence, there should be sustainable, integrated, and coordinated malaria prevention and control measures and proper water management to eliminate mosquito breeding sites [36].

On the other hand, undernutrition is one of the most serious health issues facing children in poor countries like Ethiopia, and it can have a negative impact on their physical and mental development [37, 38]. In the studied area, 54.7% (95% CI: 50.3–59.1) of children suffered from malnutrition. This result was higher than studies reported from Sekota Town in northeast Ethiopia (46.3%) [39], rural Ethiopia (48.5%) [40], and the pooled prevalence of undernutrition among underfive-year-old children in Ethiopia (49.0%) [41]. In armed internal conflict, individuals cannot commit fully to jobs due to a lack of safety, which affects families' ability to buy food [42], a key determinant of malnutrition [43]. Healthcare services are frequently in short supply during conflicts, either because medical facilities are inaccessible due to insecurity or because they become potential targets as part of war strategy [44]. Routine immunization, a major determinant of undernutrition, is one of the most affected healthcare services for children during wars [43].

According to the findings of this study, underweight (26.5%) was far more common than stunting or wasting. The prevalence of underweight was consistent with a study done in Debre Berhan Town, Ethiopia (26%) [45], and South Ari district, South Ethiopia (29.7%) [46]. In this study, the prevalence of stunting was 16.8%. This finding was higher than studies conducted in Mecha, northwest Ethiopia (11.6%) [38]. There may have been differences in sociocultural, economic, and health-related factors between the study areas that could account for the varying results. The disparities could be linked to direct causes of undernutrition such as levels of inadequate nutrition, recurrent infections, and chronic illnesses leading to reduced nutrient intake, absorption, or utilization of foods. Another potential explanation may be that poor food security and diversity in the study region are directly tied to childhood undernutrition. However, the finding was lower than previous studies conducted in Debre Berhan Town, Ethiopia (41%) [45], a study done on childhood undernutrition in Ethiopia using nationally representative data [47], as well as a systematic review and meta-analysis of undernutrition among children on ART in Ethiopia (32.98%) [48]. These variations might be due to differences in sample size and characteristics of the study participants across the studies.

This study found the prevalence of stunting to be 16.8%. Stunting is an important indicator of undernutrition that poses a major public health challenge in Ethiopia, especially among school-aged children, with the potential to impact physical and/or mental development. A notably high prevalence of stunting was observed in the study region. In response, the Ethiopian government launched the Seqota Declaration, named after the study area, pledging to eliminate child malnutrition in the country by 2030 [49]. Furthermore, 11.4% of the study participants were wasted. This was consistent with studies done in Mecha, northwest Ethiopia (10.8%) [38] and Sekota town, northeast Ethiopia (10.2%) [51]. On the other hand, this finding was lower than the finding from northeast Ethiopia (20.7%) [50], Debre Berhan Town, Ethiopia (41%) [45], and the Philippines (59.7%) [51]. This difference might be due to differences in community awareness about nutrition, the socioeconomic status of families, health service coverage, levels of poverty, the prevalence of diarrhea, and dietary diversity.

Low dietary diversity score, low meal frequency, and having a confirmed malaria infection were independent predictors of undernutrition among children in this study. A similar

association was reported between dietary diversity score in Sekota Town, Northeast Ethiopia [39] and Burkina Faso [52], and meal frequency in Dessie Town and Bahir Dar, Ethiopia [50, 53], which showed a statistically significant association. Food and nutrition insecurity becomes increasingly worse in areas affected by armed conflict. Children living in war-torn settings face a disproportionate burden of undernutrition and poor health outcomes. Therefore, there is a need for a stronger evidence-based humanitarian response to crises.

Additionally, being infected with malaria was a predictor of undernutrition, which agrees with a study conducted in Bahir Dar, Ethiopia [54]. Malaria infections can impair food and nutrient absorption by reducing appetite, increasing metabolic demands, weakening the transport of nutrients to tissues, and altering the gut lumen, leading to undernutrition [55, 56]. Moreover, severe malaria can cause dysentery. In turn, diarrhea is a significant predictor of undernutrition among children, as indicated by a study conducted in Mozambique [57]. At a population level, there is evidence that an increase in malaria infection rates was associated with peaks in admission to therapeutic feeding programs. Meanwhile, children with malnutrition also showed a greater risk of complications from malaria requiring hospitalization [58]. Therefore, efforts should prioritize implementing practical, low-cost malaria control interventions as well as detecting and treating undernutrition among vulnerable children in conflict-affected areas. It is crucial to simultaneously address the underlying dietary, vector-borne, water access, and hygiene factors that exacerbate undernutrition's impacts. A community-centered approach focusing on the root nutritional and environmental causes can help build resilience against malaria and malnutrition in sustainable, equitable ways.

## Limitation of the study

The study excluded some kebeles in the Abergele District that were experiencing ongoing conflict. This omission left out data from areas likely facing higher levels of conflict, malaria, food insecurity, and malnutrition, as vulnerable populations in those locations were not included. As a result, the study may have underestimated the true burden of malaria and undernutrition across the entire district. Furthermore, the intensity of parasite levels, micronutrient intake, and red blood cell counts in the children could not be measured due to a lack of laboratory equipment and reagents. Based on these limitations, we suggest that similar future studies should consider and aim to mitigate the types of challenges outlined above.

## Conclusion and recommendations

Both malaria and undernutrition become increasingly worse in areas affected by armed conflict. The prevalence of both malaria and undernutrition was high among children in the war-torn areas of Northeast Ethiopia. Malaria infection was significantly associated with the presence of stagnant water in residential areas, improper utilization of ITNs, an inadequate number of ITNs, and the absence of environmental management. On the other hand, dietary diversity score, meal frequency, and having a confirmed malaria infection were independent predictors of undernutrition. The results imply the need for strengthening integrated strategies to reduce both malaria and undernutrition. With the prolonged political instability in Ethiopia, sustainable malaria and nutritional interventions are necessary to alleviate difficult conditions in conflict-exposed areas. Mobile clinics and community health workers can effectively screen, treat, and monitor children for malaria and malnutrition in conflict-affected areas by integrating evaluations into flexible service models and securing ongoing international support. Improved access to all malaria and undernutrition interventions is critical for achieving the Sustainable Development Goals, but consistency across different sectors is also required.

## Supporting information

**S1 File. English version of the study questionnaire.**
(PDF)

## Acknowledgments

The authors would like to thank the study participants, data collectors, and Waghemra Zone Health Office staff for their support and unreserved cooperation in making this study a fruitful work.

## Author Contributions

**Conceptualization:** Habtu Debash, Ermiyas Alemayehu, Melaku Ashagrie Belete, Hussen Ebrahim, Ousman Mohammed, Daniel Gebretsadik, Mihret Tilahun, Agumas Shibabaw, Zewudu Mulatie, Bruktawit Eshetu, Saba Gebremichael, Alemu Gedefie.

**Data curation:** Habtu Debash, Ermiyas Alemayehu, Melaku Ashagrie Belete, Hussen Ebrahim, Ousman Mohammed, Daniel Gebretsadik, Mihret Tilahun, Agumas Shibabaw, Zewudu Mulatie, Bruktawit Eshetu, Saba Gebremichael, Alemu Gedefie.

**Formal analysis:** Habtu Debash, Ermiyas Alemayehu, Melaku Ashagrie Belete, Hussen Ebrahim, Ousman Mohammed, Daniel Gebretsadik, Mihret Tilahun, Agumas Shibabaw, Zewudu Mulatie, Bruktawit Eshetu, Saba Gebremichael, Alemu Gedefie.

**Funding acquisition:** Habtu Debash, Ermiyas Alemayehu, Melaku Ashagrie Belete, Hussen Ebrahim, Ousman Mohammed, Daniel Gebretsadik, Mihret Tilahun, Agumas Shibabaw, Zewudu Mulatie, Bruktawit Eshetu, Saba Gebremichael, Alemu Gedefie.

**Investigation:** Habtu Debash, Ermiyas Alemayehu, Melaku Ashagrie Belete, Hussen Ebrahim, Ousman Mohammed, Daniel Gebretsadik, Mihret Tilahun, Agumas Shibabaw, Zewudu Mulatie, Bruktawit Eshetu, Saba Gebremichael, Alemu Gedefie.

**Methodology:** Habtu Debash, Ermiyas Alemayehu, Melaku Ashagrie Belete, Hussen Ebrahim, Ousman Mohammed, Daniel Gebretsadik, Mihret Tilahun, Agumas Shibabaw, Zewudu Mulatie, Bruktawit Eshetu, Saba Gebremichael, Alemu Gedefie.

**Project administration:** Habtu Debash, Ermiyas Alemayehu, Melaku Ashagrie Belete, Hussen Ebrahim, Ousman Mohammed, Daniel Gebretsadik, Mihret Tilahun, Agumas Shibabaw, Zewudu Mulatie, Bruktawit Eshetu, Saba Gebremichael, Alemu Gedefie.

**Resources:** Habtu Debash, Ermiyas Alemayehu, Melaku Ashagrie Belete, Hussen Ebrahim, Ousman Mohammed, Daniel Gebretsadik, Mihret Tilahun, Agumas Shibabaw, Zewudu Mulatie, Bruktawit Eshetu, Saba Gebremichael, Alemu Gedefie.

**Software:** Habtu Debash, Ermiyas Alemayehu, Melaku Ashagrie Belete, Hussen Ebrahim, Ousman Mohammed, Daniel Gebretsadik, Mihret Tilahun, Agumas Shibabaw, Zewudu Mulatie, Bruktawit Eshetu, Saba Gebremichael, Alemu Gedefie.

**Supervision:** Habtu Debash, Ermiyas Alemayehu, Melaku Ashagrie Belete, Hussen Ebrahim, Ousman Mohammed, Daniel Gebretsadik, Mihret Tilahun, Agumas Shibabaw, Zewudu Mulatie, Bruktawit Eshetu, Saba Gebremichael, Alemu Gedefie.

**Validation:** Habtu Debash, Ermiyas Alemayehu, Melaku Ashagrie Belete, Hussen Ebrahim, Ousman Mohammed, Daniel Gebretsadik, Mihret Tilahun, Agumas Shibabaw, Zewudu Mulatie, Bruktawit Eshetu, Saba Gebremichael, Alemu Gedefie.

**Visualization:** Habtu Debash, Ermiyas Alemayehu, Melaku Ashagrie Belete, Hussen Ebrahim, Ousman Mohammed, Daniel Gebretsadik, Mihret Tilahun, Agumas Shibabaw, Zewudu Mulatie, Bruktawit Eshetu, Saba Gebremichael, Alemu Gedefie.

**Writing – original draft:** Habtu Debash.

**Writing – review & editing:** Habtu Debash, Ermiyas Alemayehu, Melaku Ashagrie Belete, Hussen Ebrahim, Alemu Gedefie.

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
