## [Decision Letter · Decision Letter 0]

15 Aug 2024

PONE-D-24-27373Magnitude and Determinants of Malaria and Undernutrition among Acutely Febrile Children Visiting Temporary Malaria Screening Sites in War-Torn Areas of Northeast EthiopiaPLOS ONE

Dear Dr. Debash,

Thank you for submitting your manuscript to PLOS ONE. After careful consideration, we felt that your study has the potential to be published if it is revised to address specific topics raised by the reviewers. According to the reviewers, there are several specific topics where further improvements would be of substantial benefit to readers; for example, A detailed description of the statistical analyzes is necessary, including strategies for selecting the variables of interest and adjusting the multivariate models. For your guidance, a copy of the reviewers' comments was included below.  

We look forward to receiving your revised manuscript.

Kind regards,

Luzia H Carvalho, Ph.D.

Academic Editor

PLOS ONE

Journal Requirements:

2. In the online submission form, you indicated that All data generated in this study are included in the manuscript. The data sets analysed during the current study are available from the corresponding author on reasonable request.

Reviewers' comments:

Reviewer's Responses to Questions

**Comments to the Author**

1. Is the manuscript technically sound, and do the data support the conclusions?

Reviewer #1: Yes

Reviewer #2: Yes

2. Has the statistical analysis been performed appropriately and rigorously? 

Reviewer #1: Yes

Reviewer #2: Yes

3. Have the authors made all data underlying the findings in their manuscript fully available?

Reviewer #1: Yes

Reviewer #2: Yes

4. Is the manuscript presented in an intelligible fashion and written in standard English?

Reviewer #1: Yes

Reviewer #2: Yes

5. Review Comments to the Author

**Reviewer #1:** This cross-sectional analysis considers the prevalence and correlates of malaria and undernutrition in a conflict-affected area of Ethiopia. The study design and analysis are appropriate and data are adequately presented. My comments are minor.

**Reviewer #2:** Comments and Suggestions for Manuscript Revision

Manuscript Number: PONE-D-24-27373

Manuscript Title: Magnitude and Determinants of Malaria and Undernutrition among Acutely Febrile Children Visiting Temporary Malaria Screening Sites in War-Torn Areas of Northeast Ethiopia

Introduction

Justification and contextualization: The introduction clearly presents the relevance of the study, highlighting the importance of understanding the prevalence and risk factors of malaria and undernutrition in conflict areas.

Methodology

Study design: The study is cross-sectional, conducted between November 2022 and January 2023. The use of systematic random sampling is appropriate for this type of study.

Data collection: The data collection via questionnaire, capillary blood samples, and anthropometric measurements is well described. However, it would be useful to include information about the validation of the questionnaire used.

Statistical analysis: The use of bivariate and multivariate logistic regression is appropriate. However, it is essential to ensure that all confounding variables are adequately controlled.

o Suggestion: Include a detailed description of the statistical analysis, including the variable selection strategy (stepwise) and how the adjustment of the multivariate models was performed, including the magnitude of the effect after the exponentiation of the beta from the logistic regression model.

Results

Prevalence of malaria and undernutrition: The results show a high prevalence of malaria (65.9%) and undernutrition (54.7%). It is important to discuss whether these results are consistent with other studies in similar contexts.

o Suggestion 1: Simplify the description of the results to avoid excessive repetition and make the reading more fluid. It is not necessary to repeat results for all categories and all numbers that are in the table.

o

o Suggestion 2: Consolidate Table 3 into Table 2 to avoid redundancy and facilitate reading.

o Suggestion 3: The column for the total tested in Table 2 is redundant to Table 1. I recommend removing it, as this information is already included in Table 1.

2. Analysis of risk factors: The independent factors associated with malaria include the presence of stagnant water, inadequate use of mosquito nets, and lack of environmental management. For undernutrition, dietary diversity and meal frequency are highlighted as significant factors.

Discussion

Comparison with other studies: The comparison with other studies is made but could be expanded to include more details on why certain prevalences are higher or lower compared to other regions.

o Suggestion: Include an explanation of why diarrhea lost significant association after adjustment in the multivariate model and cite relevant studies, as it is a significant predictor in undernutrition. I suggest citing the article "Predictors of nutritional recovery time in children aged 6–59 months with severe acute malnutrition in Sofala Province, Mozambique: survival analysis approach" by Audêncio Victor et al., published in the Journal of Public Health. This study can be accessed at: https://doi.org/10.1093/pubmed/fdae049.

Implications of the results: The discussion addresses well the implications of the findings for public health. However, it would be useful to include more concrete suggestions on how interventions can be implemented in conflict areas.

Study limitations: The limitations are mentioned but could be discussed in more detail, especially regarding potential selection biases and data collection limitations.

Conclusion

Summarize the main findings: The conclusion adequately summarizes the main findings of the study and suggests the need for integrated strategies to combat malaria and undernutrition in conflict regions.

Structure and Clarity

Manuscript Organization: The manuscript is well-organized, but the introduction could be more concise.

Language and Grammar: The language is clear, but additional grammatical revision may be needed to ensure fluency.

Ethical Considerations

Ethical Approval: Obtaining ethical approval and informed consent is well documented, which is essential for publication.

Recommendations for the Authors

Improvement in Methodology Description: Include more details about the validation of the questionnaire and control of confounding variables.

Deeper discussion of limitations: Expand the limitations section to include more details about possible biases and limitations of the study.

Expansion of discussion on interventions: Provide more concrete and detailed suggestions on how interventions can be implemented and evaluated in conflict contexts.

6. PLOS authors have the option to publish the peer review history of their article (what does this mean?). If published, this will include your full peer review and any attached files.

Reviewer #1: No

Reviewer #2: No

---

## [Author Response · Author response to Decision Letter 0]

31 Aug 2024

Point by point response to reviewers’ and editors’ comments

Dear Editor and reviewers

Thank you very much for your positive feedback to our work. We revised the manuscript based on comments raised and corrections we made indicated in track change file. Thank you.

Manuscript Title: Magnitude and Determinants of Malaria and Undernutrition among Acutely Febrile Children Visiting Temporary Malaria Screening Sites in War-Torn Areas of Northeast Ethiopia

Response to reviewer #1 comments

Abstract

1. 422---This is not approximate. It is the actual number of children.

Response: thank you, we corrected it.

2. Where were they randomly selected from? Can the data be considered representative of any area / population?

Response: Febrile children were selected using a systematic random sampling technique from febrile children attending temporary malaria screening sites in Abergelie district. Regarding to the representation of the area/population we indicated in the revised manuscript. Thank you. 

3. Stadiometer? 

Response: stadiometer was not available in the settings, so, we used meter to measure heights of children.

4. The word "determinants" is misused here and elsewhere in the manuscript. It is not a determinant until there is a statistically significant association and, in a cross-sectional study, I would still be hesitant to call it a determinant even with that. It could be called a correlate or a factor.

Response: Thank you for your constructive comment, we amend it as correlates.

5. You would generally include data here, at the very least P values. At least indicate directionality.

Response: we included P-values.

Introduction

6. or it may be that the citation at the end of the paragraph needs to be moved up

Response: we indicated reference 3.

7. You use "malnutrition" elsewhere and sometimes "under-nutrition." I recommend sticking with undernutrition throughout and treating it as one word.

Response: we corrected it and consistently used the word undernutrition.

8. Temporal?

Response: we replaced by temporary. Thank you.

Methods

9. Abergelle vs Abergele?

Response: Abergele is the correct one, so we correct it. 

10. In inclusion and exclusion criteria; presented with?, children's, critical ill-How was this defined?

Response: we amended and corrected them.

11. Anthropometric measurement; The software is used to derive anthropometric indices. You use the cut-offs to calculate indicators of undernutrition.

Response: The software calculates anthropometric indices and uses established cut-offs to identify indicators of undernutrition effectively.

12. Errors in table 1, Wasn't this by design? If so, is it needed in the table? 

Response: we made changes. Thank you.

13. Are these proportions different? It seems like it's the column proportion in the "total BF examed" but the rest are rows? It's confusing. Need to include the abbreviation for Bf

Response: we merged and amend table 2 and 3, made changes, the the relative proportion of Plasmodium species from the total cases expressed in their rows. 

14. Same comment as for the p-values. There is no need for three digits after the decimal.

Response: we corrected them accordingly.

15. This is an unusual way to present undernutrition. It also seems strange that there are no underweight stunted children based on the way that wasted + stunted + underweight add precisely to 231.

Response: Thank you for your insightful feedback regarding the presentation of undernutrition in our study. The categorization of undernutrition in our study follows WHO definitions, which include underweight, stunting, and wasting. The absence of overlapping undernutrition children may seem unusual, but it highlights that not all stunted children are underweight. This can occur due to various factors affecting growth and weight differently. In fact, a child can be stunted due to chronic malnutrition without being underweight at a given point in time. Thank you.

Discussion

16. All of the studies cited in these two paragraphs comparable? It seems like you would expected a higher prevalence of malaria when you are specifically measuring it in children presenting with malaria-like symptoms at malaria surveillance sites. If these studies did not have a comparable design, I do not think that it adds anything to the discussion to compare your findings with these from other sites.

Response: thank you so much we delete it.

17. It is unclear why data from the Philippines are cited here. It's not relevant. Those small-area surveys from within Ethiopia may be relevant, or something from the DHS.

Response: we amend this paragraph and correct using relevant literatures. Thank you.

Response to reviewer #2 comments

1. Data collection: The data collection via questionnaire, capillary blood samples, and anthropometric measurements is well described. However, it would be useful to include information about the validation of the questionnaire used.

Response: we included information about the validation of the questionnaire we used in the revised manuscript.

2. Statistical analysis: The use of bivariate and multivariate logistic regression is appropriate. However, it is essential to ensure that all confounding variables are adequately controlled. Suggestion: Include a detailed description of the statistical analysis, including the variable selection strategy (stepwise) and how the adjustment of the multivariate models was performed, including the magnitude of the effect after the exponentiation of the beta from the logistic regression model.

3. Response: we incorporated the comments in this revised manuscript. Thank you.

Results

4. Prevalence of malaria and undernutrition: The results show a high prevalence of malaria (65.9%) and undernutrition (54.7%). It is important to discuss whether these results are consistent with other studies in similar contexts.

o Suggestion 1: Simplify the description of the results to avoid excessive repetition and make the reading more fluid. It is not necessary to repeat results for all categories and all numbers that are in the table.

o Response: Based on this comment, we have minimized the description of tables to avoid excessive repetition.

o Suggestion 2: Consolidate Table 3 into Table 2 to avoid redundancy and facilitate reading.

o Response: we merged them 

o Suggestion 3: The column for the total tested in Table 2 is redundant to Table 1. I recommend removing it, as this information is already included in Table 1.

o Response: we delete it. Thank you

Discussion

5. Comparison with other studies: The comparison with other studies is made but could be expanded to include more details on why certain prevalences are higher or lower compared to other regions.

Response: Thank you for your positive feedback. We have tried to expand on the comparable results by adding their justification.

o Suggestion: Include an explanation of why diarrhea lost significant association after adjustment in the multivariate model and cite relevant studies, as it is a significant predictor in undernutrition. I suggest citing the article "Predictors of nutritional recovery time in children aged 6–59 months with severe acute malnutrition in Sofala Province, Mozambique: survival analysis approach" by Audêncio Victor et al., published in the Journal of Public Health. This study can be accessed at: https://doi.org/10.1093/pubmed/fdae049.

o Response: we included the above article in the revised manuscript.

6. Implications of the results: The discussion addresses well the implications of the findings for public health. However, it would be useful to include more concrete suggestions on how interventions can be implemented in conflict areas.

Response: thank you for your feedback; we have tried to include the above comment.

7. Study limitations: The limitations are mentioned but could be discussed in more detail, especially regarding potential selection biases and data collection limitations.

Response: we included the limitation of the study in the revised document and indicted the limitations in the revised manuscript.

Conclusion

8. Summarize the main findings: The conclusion adequately summarizes the main findings of the study and suggests the need for integrated strategies to combat malaria and undernutrition in conflict regions.

Response: thank you.

Structure and Clarity

9. Manuscript Organization: The manuscript is well-organized, but the introduction could be more concise.

Response: In this revised manuscript, we have made the introduction section more organized.

10. Language and Grammar: The language is clear, but additional grammatical revision may be needed to ensure fluency.

Response: We made grammatical revisions throughout the document to improve it.

11. Ethical Considerations, Ethical Approval: Obtaining ethical approval and informed consent is well documented, which is essential for publication.

Response: thank you.

Recommendations for the Authors

12. Improvement in Methodology Description: Include more details about the validation of the questionnaire and control of confounding variables.

Response: thank you, we stated in the revised manuscript.

13. Deeper discussion of limitations: Expand the limitations section to include more details about possible biases and limitations of the study.

Response: we expand the limitation of the study in the revised version

14. Expansion of discussion on interventions: Provide more concrete and detailed suggestions on how interventions can be implemented and evaluated in conflict contexts.

Response: we have tried to make concrete. Thank you again.

---

## [Decision Letter · Decision Letter 1]

13 Sep 2024

PONE-D-24-27373R1Prevalence and Correlates of Malaria and Undernutrition among Acutely Febrile Children Visiting Temporary Malaria Screening Sites in War-Torn Areas of Northeast EthiopiaPLOS ONE

Dear Dr. Debash,

Thank you for submitting your manuscript to PLoS ONE. After careful consideration, we feel that your manuscript will likely be suitable for publication if the authors revise it to address a specific point raised   by the reviewer. According to the reviewer, a professional copy-editing service  would be of substantial benefit to the readers.   For your guidance, a copy of the reviewers' comments was included below.  

We look forward to receiving your revised manuscript.

Kind regards,

Luzia H Carvalho, Ph.D.

Academic Editor

PLOS ONE

Journal Requirements:

Reviewers' comments:

Reviewer's Responses to Questions

**Comments to the Author**

1. If the authors have adequately addressed your comments raised in a previous round of review and you feel that this manuscript is now acceptable for publication, you may indicate that here to bypass the “Comments to the Author” section, enter your conflict of interest statement in the “Confidential to Editor” section, and submit your "Accept" recommendation.

Reviewer #1: All comments have been addressed

Reviewer #2: All comments have been addressed

2. Is the manuscript technically sound, and do the data support the conclusions?

Reviewer #1: Yes

Reviewer #2: Yes

3. Has the statistical analysis been performed appropriately and rigorously? 

Reviewer #1: Yes

Reviewer #2: Yes

4. Have the authors made all data underlying the findings in their manuscript fully available?

Reviewer #1: Yes

Reviewer #2: Yes

5. Is the manuscript presented in an intelligible fashion and written in standard English?

Reviewer #1: Yes

Reviewer #2: Yes

6. Review Comments to the Author

Reviewer #1: The authors have thoroughly responded to all of the feedback from both reviewers. I have no further comments on the manuscript.

Reviewer #2: Comments to the Author:

Congratulations on thoroughly addressing all the previous suggestions and comments. The revised manuscript is significantly improved in terms of clarity and methodological precision. Below are my final comments:

Technical Soundness: The study is robust, and the data appropriately support the conclusions. The use of bivariate and multivariate logistic regression was applied correctly, and confounding variables were adequately controlled. The results section is clear and concise.

Statistical Analysis: The adjustments made to the statistical analyses, including a more detailed description of the variable selection in the model, were excellent additions that strengthen the interpretation of the data.

Data Availability: The author has ensured that all data are available in accordance with PLOS ONE’s policies, facilitating reproducibility.

Clarity and Grammar: The manuscript is written in clear and intelligible English. However, minor grammatical improvements could further polish the text, though this does not impede the acceptance of the manuscript.

Suggestions:

The comparison with other studies has been expanded and now includes clearer justifications for the observed differences in prevalence, which enriches the discussion.

The section on study limitations has been improved, offering detailed insights into potential biases, especially regarding the exclusion of certain areas due to conflict.

I suggest a final grammatical revision to ensure the fluency of the text is impeccable.

Overall, all my previous concerns have been satisfactorily addressed, and I see no further obstacles to accepting this manuscript for publication.

7. PLOS authors have the option to publish the peer review history of their article (what does this mean?). If published, this will include your full peer review and any attached files.

Reviewer #1: No

Reviewer #2: **Yes: **Audêncio Victor

---

## [Author Response · Author response to Decision Letter 1]

19 Sep 2024

Point by point response to reviewers’ and editors’ comments

Dear editor and reviewers. 

Thank you very much for your positive feedback on our work. We greatly appreciate you taking the time to review our manuscript. Based on the comments raised by the reviewers, we revised the manuscript and indicated all changes made in a tracked changes file. Your acknowledgment that we have adequately addressed all points raised through our revisions is encouraging. Thank you again for your feedback and consideration of our manuscript for potential publication.

Manuscript Title: Magnitude and Determinants of Malaria and Undernutrition among Acutely Febrile Children Visiting Temporary Malaria Screening Sites in War-Torn Areas of Northeast Ethiopia

Response to reviewer #1 

Comment #1: The authors have thoroughly responded to all of the feedback from both reviewers. I have no further comments on the manuscript.

Response: Thank you for taking the time to evaluate our manuscript and provide feedback. We appreciate you acknowledging that we have addressed all the points raised by the reviewer. Your recommendation that the manuscript requires no further revision is helpful and an important step before a decision on publication can be made. Your review has strengthened our work by confirming that we have adequately responded to the comments and suggestions for improvement. We believe the manuscript is now suitably revised for consideration for publication. Thank you again for your thoughtful review.

Response to reviewer #2 

Reviewer #2: Comments to the Author:

Congratulations on thoroughly addressing all the previous suggestions and comments. The revised manuscript is significantly improved in terms of clarity and methodological precision. Below are my final comments:

Comment Suggestions:

The comparison with other studies has been expanded and now includes clearer justifications for the observed differences in prevalence, which enriches the discussion.

The section on study limitations has been improved, offering detailed insights into potential biases, especially regarding the exclusion of certain areas due to conflict.

I suggest a final grammatical revision to ensure the fluency of the text is impeccable.

Overall, all my previous concerns have been satisfactorily addressed, and I see no further obstacles to accepting this manuscript for publication.

Response: Thank you for your thorough review and helpful feedback on our revised manuscript. We are glad to hear that you feel we have satisfactorily addressed your previous concerns. We appreciate you noting that the expansion of the comparison with other studies has strengthened the discussion by including clearer justifications for differences observed. Additionally, your comments regarding our improved section on study limitations and the provision of detailed insights into potential biases are greatly valued. 

As per your suggestion, we have conducted a final grammatical revision to ensure textual fluency is impeccable. We believe the manuscript is now rigorous and well-rounded as a result of incorporating the changes recommended by you and the other reviewer. Thank you for confirming there are no further obstacles to accepting our work for publication. We feel privileged to contribute our research to the journal under your guidance. Thank you again.

---

## [Editor Report · Decision Letter 2]

27 Sep 2024

Prevalence and Correlates of Malaria and Undernutrition among Acutely Febrile Children Visiting Temporary Malaria Screening Sites in War-Torn Areas of Northeast Ethiopia

PONE-D-24-27373R2

Dear Dr. Debash,

We’re pleased to inform you that your manuscript has been judged scientifically suitable for publication and will be formally accepted for publication once it meets all outstanding technical requirements.

Kind regards,

Luzia H Carvalho, Ph.D.

Academic Editor

PLOS ONE
---

## [Editor Report · Acceptance letter]

8 Oct 2024

PONE-D-24-27373R2 

PLOS ONE

Dear Dr. Debash, 

I'm pleased to inform you that your manuscript has been deemed suitable for publication in PLOS ONE. Congratulations! Your manuscript is now being handed over to our production team.

Kind regards, 

on behalf of

Dr. Luzia H Carvalho 

Academic Editor

PLOS ONE